# Diagnostic accuracy of adenosine deaminase for pleural tuberculosis in a low prevalence setting: A machine learning approach within a 7-year prospective multi-center study

Alberto Garcia-Zamalloa[1,2]*, Diego Vicente[3,4], Rafael Arnay[5], Arantzazu Arrospide[6,7,8,9], Jorge Taboada[10], Iván Castilla-Rodríguez[5,9], Urko Aguirre[8,9,11], Nekane Múgica[12], Ladislao Aldama[12], Borja Aguinagalde[13], Montserrat Jimenez[14], Edurne Bikuña[14], Miren Begoña Basauri[15], Marta Alonso[3], Emilio Perez-Trallero[3], with the Gipuzkoa Pleura Group Consortium¶

1 Internal Medicine Service, Osakidetza/Basque Health Service, Mendaro Hospital, Gipuzkoa, Spain, 2 Mycobacterial Infection Study Group (GEIM), From the Spanish Infectious Diseases Society, Spain, 3 Microbiology Department, Respiratory Infection and Antimicrobial Resistance Group. Osakidetza/Basque Health Service, Biodonostia Health Research Institute, Donostia University Hospital, Gipuzkoa, Spain, 4 Faculty of Medicine, University of the Basque Country, UPV/EHU, Gipuzkoa, Donostia, Spain, 5 Departamento de Ingeniería Informática y de Sistemas, Universidad de La Laguna, Santa Cruz de Tenerife, Spain, 6 Gipuzkoa Primary Care-Integrated Health Organisation Research Unit, Osakidetza/Basque Health Service, Debagoiena Integrated Health Organisation, Alto Deba Hospital, Arrasate-Mondragon, Spain, 7 Epidemiology and Public Health Area, Economic Evaluation of Chronic Diseases Research Group, Biodonostia Health Research Institute, Donostia, Spain, 8 Kronikgune Institute for Health Services Research, Bizkaia/Barakaldo, Spain, 9 Health Services Research on Chronic Patients Network (REDISSEC), Spain, 10 Preventive Medicine and Western Gipuzkoa Clinical Research Unit, Osakidetza/Basque Health Service, Mendaro Hospital, Gipuzkoa, Spain, 11 Osakidetza/Basque Health Service, Research Unit, Galdakao University Hospital, Bizkaia, Spain, 12 Pneumology Service, Osakidetza/Basque Health Service, Donostia University Hospital, Gipuzkoa. Spain, 13 Thoracic Surgery Service, Osakidetza/Basque Health Service, Donostia University Hospital, Gipuzkoa, Spain, 14 Epidemiological Surveillance Unit, Health Department, Basque Government, Gipuzkoa, Spain, 15 Biochemistry Laboratory, Osakidetza/Basque Health Service, Mendaro Hospital, Gipuzkoa, Spain

¶ Membership list can be listed in the Acknowledgments section.
* alberto.garciazamalloa@gmail.com

## Abstract

### Objective

To analyze the performance of adenosine deaminase in pleural fluid combined with other parameters routinely measured in clinical practice and assisted by machine learning algorithms for the diagnosis of pleural tuberculosis in a low prevalence setting, and secondly, to identify effusions that are non-tuberculous and most likely malignant.

### Patients and methods

We prospectively analyzed 230 consecutive patients diagnosed with lymphocytic exudative pleural effusion from March 2013 to June 2020. Diagnosis according to the composite reference standard was achieved in all cases. Pre-test probability of pleural tuberculosis was 3.8% throughout the study period. Parameters included were: levels of adenosine

**Data Availability Statement:** All relevant data are within the manuscript and its Supporting information files.

**Funding:** As we state in the manuscript, our study was supported by grant PI2016111036 to Dr. Diego Vicente from the Office of Health Research and Innovation of the Department of Health of the Basque Government. Funding was mainly used for payment of Xpert MTB/RIF in pleural fluid and tissue samples. There was no role of the funders regarding the study design, data collection, medical decisions or preparation of the manuscript.

**Competing interests:** The authors have declared that no competing interests exist.

deaminase, pH, glucose, proteins, and lactate dehydrogenase, red and white cell counts and lymphocyte percentage in pleural fluid, as well as age. We tested six different machine learning-based classifiers to categorize the patients. Two different classifications were performed: a) tuberculous/non-tuberculous and b) tuberculous/malignant/other.

## Results

Out of a total of 230 patients with pleural effusion included in the study, 124 were diagnosed with malignant effusion and 44 with pleural tuberculosis, while 62 were given other diagnoses. In the tuberculous/non-tuberculous classification, and taking into account the validation predictions, the support vector machine yielded the best result: an AUC of 0.98, accuracy of 97%, sensitivity of 91%, and specificity of 98%, whilst in the tuberculous/malignant/other classification, this type of classifier yielded an overall accuracy of 80%. With this three-class classifier, the same sensitivity and specificity was achieved in the tuberculous/other classification, but it also allowed the correct classification of 90% of malignant cases.

## Conclusion

The level of adenosine deaminase in pleural fluid together with cell count, other routine biochemical parameters and age, combined with a machine-learning approach, is suitable for the diagnosis of pleural tuberculosis in a low prevalence scenario. Secondly, non-tuberculous effusions that are suspected to be malignant may also be identified with adequate accuracy.

## Introduction

Tuberculosis remains a major global public health problem: in 2019, 10 million people developed tuberculosis and it caused 1.5 million deaths, with around 5.4% of the cases diagnosed in the World Health Organization (WHO) European region and WHO region of the Americas [1]. Extrapulmonary disease is the initial presentation in about 25% of patients, involving lymph nodes and primarily the pleura [2]; as a result, tuberculosis is currently one of the most frequent causes of exudative pleural effusion worldwide [3]. Tuberculous pleural effusion (TPE) is a paucibacillary manifestation of tuberculosis, and bacteriological tests in pleural fluid have historically given suboptimal results, leading to a search for biomarkers in pleural fluid and the use of aggressive diagnostic techniques like pleural biopsy [4].

Adenosine deaminase (ADA) has proven to be the most useful and cost-effective biomarker in pleural fluid, but based on the Bayesian interpretation of its diagnostic accuracy, it is currently only accepted as a rule-out test in low prevalence scenarios. The most accepted cut-off value is 40 U/l [5–7]. In 2012, we reported a retrospective study in a local scenario of decreasing incidence from 1998 to 2008, and showed that combining ADA>40 U/l and lymphocyte percentage >50% in pleural fluid the diagnostic accuracy of the former increased substantially, especially in low-to-intermediate incidence settings [8]. We decided to conduct a multicenter prospective study with diagnosis according to the composite reference standard in all patients included, in a low prevalence scenario and with 10-fold larger population.

Using the data collected in this study, we aimed to determine the most accurate machine learning algorithm for the classification of patients (a) according to the presence of tuberculosis and (b) into three categories: tuberculous, malignant or other. Machine Learning is the

science of giving computers the ability to *learn from data*. A machine learning system is trained rather than explicitly programmed. That is, such a system is presented with many examples relevant to a task, and it finds a statistical structure in these examples that, eventually, allows it to come up with rules for automating the task [9]. Machine learning algorithms constitute a powerful tool which are becoming useful in clinical medicine for the integration of multiple variables and subsequent estimation of the likelihood of a given diagnosis [10, 11].

## Materials and methods

### Data source

This prospective observational study was conducted in one tertiary hospital (Donostia University Hospital), three regional hospitals (Deba Behea Hospital, Goi Urola Hospital, Bidasoa Hospital), and two private health clinics (Gipuzkoa Polyclinic and Asuncion Clinic), offering a total of approximately 1,725 beds and all located in the province of Gipuzkoa, Basque Country, Spain, with a population of 720,000.

The mean annual incidence rate of tuberculosis in Gipuzkoa from 2013 to 2020 was 14 cases per 100,000 population (and it remained under 20 cases per 100,000 throughout this period) [12]. Data collection has been highly reliable in Gipuzkoa since 1995, and a Tuberculosis Control Program was implemented across the entire region of the Basque Country in 2003 [13].

For the purpose of this study and following the approach of previous authors since the nineteen-nineties, the term "prevalence" was used to refer to the number of cases of a specific type of pleural effusion divided by the total number of pleural effusions studied in a given clinical setting: in other words, the "pre-test probability" [14, 15]. A prevalence of less than 10% is considered low [16]. Overall, 1,177 cases of pleural effusion were diagnosed in Gipuzkoa throughout the period 2013–2020, and 45 of them were tuberculous; hence, the local prevalence of TPE was 3.8%.

Due to the fact that virtually all cases of TPE become exudative and lymphocytic [17–19], and given our limited funding and the high cost of some tests (e.g., the Xpert MTB/RIF polymerase chain reaction [PCR] test for *Mycobacterium tuberculosis*—MTB -, which was performed in all cases), we decided to include only lymphocytic exudative pleural fluid samples in this study. During the whole study period, the first pleural fluid sample was only exudative and neutrophilic in 1 out of the 45 patients diagnosed with TPE; his case was included in the prevalence calculation, but his data were not included in the rest of the analysis.

We considered it appropriate to conduct this prospective study following a PIRTO strategy:

- P (Population): patients aged 16 years or older diagnosed with lymphocytic exudative pleural effusion from March 2013 to June 2020 in a low prevalence setting like ours

- I (Index Test): ADA level in pleural fluid, combined with patients´ age and routine pleural fluid parameters obtained from the first pleural fluid sample, and using machine learning algorithms

- R (Reference test): diagnosis according to the composite reference standard for TPE, that is, culture positive for MTB or Xpert MTB/RIF assay positive for MTB in pleural fluid, pleural tissue or sputum; or granulomatous inflammation in pleural tissue

- T (Target condition): diagnostic accuracy of the aforementioned parameters for TPE in a low prevalence setting

- O (Outcome): results obtained, in terms of sensitivity, specificity, and area under the receiver operating characteristic curve (AUC) from two different machine learning approaches: tuberculous/other and tuberculous/malignant/other.

We report the above items by following the STARD guidelines [20]. Inclusion criteria in the study were: 1) age over 16 years, 2) *de novo* diagnosis of pleural effusion that is both exudative (according to Light´s criteria [21]) and lymphocytic (defined as a lymphocytes accounting for more than 50% of nucleated cells in pleural fluid [2]) in nature, and 3) etiology of the effusion determined according to the composite reference standard. Patients were not included in the study if they had a history of any disease that could explain the development of a pleural effusion (e.g., patients previously diagnosed with lung cancer who developed pleural effusion).

Further, patients in whom no definitive diagnosis was reached were not included in the study. When more than one thoracocentesis was performed, only the results (routine biochemical parameters, cell count and ADA level) from the first pleural fluid sample were included in the statistical analysis, but all samples were taken into account for obtaining the gold standard diagnosis of every patient (e.g., positive culture or malignant cells in a subsequent sample).

Through the whole period of study all the clinical information and reference standard results were available to all the members of the Consortium.

All patients gave written informed consent to their inclusion in the study before diagnostic procedures were performed. Only one patient was a minor. She was seventeen years old and her parents signed the informed consent. The protocol was evaluated and approved by the Clinical Research Ethics Committee of Euskadi (Record number 11/12).

All included patients reported demographic data (age, sex and nationality) and underwent the following types of testing:

- Pleural fluid tests including measurement of pH, glucose, protein, lactate dehydrogenase, and ADA levels, total and differential red and white cell counts, cytology, aerobic and anaerobic cultures, Lowenstein-Jensen medium and BACTEC MGIT (BACTEC Mycobacteria Growth Indicator Tube 960 System; BD Diagnostic Instrument Systems, Sparks, Maryland, USA) cultures, and Xpert MTB/RIF assays for MTB, as well as assessment of serum glucose, protein and LDH levels

- Aerobic, Lowenstein-Jensen medium and BACTEC MGIT cultures, along with cytology, of any spontaneous or induced sputum, when it was possible to obtain such samples

- Chest computed tomography (CT)

- Screening for human immunodeficiency virus (HIV), in line with the Tuberculosis Control Program for all patients diagnosed with tuberculosis implemented in the Basque Country since 2003 [13]

- Fibrobronchoscopy, depending on the CT findings, samples taken being sent for histopathological examination, aerobic and anaerobic cultures, Lowenstein-Jensen and BACTEC MGIT cultures, as well as Xpert MTB/RIF assays for MTB

- Pleural biopsy (either closed or thoracoscopic), when necessary to reach a diagnosis, samples taken being sent for histopathological exam examination, aerobic and anaerobic cultures, and Lowenstein-Jensen and BACTEC MGIT cultures, as well as Xpert MTB/RIF assays for MTB.

Pleural fluid ADA level was measured using an automated ultraviolet kinetic assay (Roche Diagnostics, Mannheim, Germany). White blood cell count was obtained with a automated

hematology analyzer (Sysmex XN-1000, Roche Diagnostics), and a differential count was performed manually, after Wright staining. Mycobacterial cultures were performed on Lowenstein-Jensen medium and also in MGIT 960 culture tubes.

Regarding diagnostic criteria, patients were classified as having 1) tuberculous pleural effusion when cultures or Xpert MTB/RIF assays for MTB in pleural fluid, pleural biopsy or sputum were positive, or when granulomatous inflammation was evident in pleural tissue; 2) malignant pleural effusion (MPE) when malignant cells were found in pleural fluid or pleural biopsy tissue, or paramalignant pleural effusion when cancer was diagnosed *de novo*, no other cause of pleural effusion was identified and the pleural effusion and tumour were concurrent and developed in parallel, but no malignant cells were demonstrated in the pleural fluid or tissue [22, 23]; 3) parapneumonic effusion when there was any exudative pleural effusion associated with clinical and radiological pneumonia and complete clinical recovery and resolution of abnormal findings with antibiotics as the only treatment; and 4) miscellaneous effusion when they had any other type of pleural effusion with another aetiology and defined by well-established clinical criteria.

Calculation of sample size was performed by using a statistical program (Epidat 3.1). For a sensitivity of 95%, specificity of 90%, prevalence (pre-test probability) of 10%, significance level of 5%, power of 80% and precision of 5%, the minimum sample size was 200 patients. Since a 25% loss was estimated, final sample size was set to 250 patients.

Fig 1 shows a flowchart depicting the patient selection process.

## Statistical analysis

With the data collected from the patients included, two different analyses were performed. The main analysis used machine learning techniques to classify samples in two categories: tuberculous and non-tuberculous. A secondary analysis used the same techniques to additionally distinguish malignant cases, yielding three categories: tuberculous, malignant and "other".

Both analyses split the data into a training set (80%) and test set (20%), to maintain the proportion of the three classes in both sets (19.1% tuberculous, 53.9% malignant and 26.9% other). The training and testing processes used the following features: age, ADA level and routine parameters from pleural fluid (pH, glucose, protein, and lactate dehydrogenase levels and total and differential red and white cell counts). S1 Table shows an estimate of the relative importance of these features in the classification process. We carried out a data standardization process before training: for each feature in the training set, we first subtracted the mean value (such that standardized values always had a zero mean), and then divided by the standard deviation (such that the resulting distribution had unit variance). The mean value and standard deviation calculated in the training set were also used in the standardization process of the test set.

As we did not have a priori knowledge about what type of classifier would work best in the proposed problem, six of the most widely used were trained and tested: multi layer perceptron, logistic regression, support vector machine, decision tree, K-nearest neighbors and random forest. We used the Python scikit-learn implementation of these classifiers [24]. A set of different values was considered for each of the classifier parameters [25] in combination with a 5-fold cross validation [26], to determine the combination of parameters that optimized a given metric. In the case of the classification between tuberculous and non-tuberculous, a binary classification problem, we used accuracy, the AUC and the F1 score, which is the harmonic mean of precision and recall [27]. The AUC led to the best parameterizations in this case. In the classification between tuberculous, malignant and other effusions, this being a multiclass classification problem, we used different metrics, namely, the weighted F1 score,

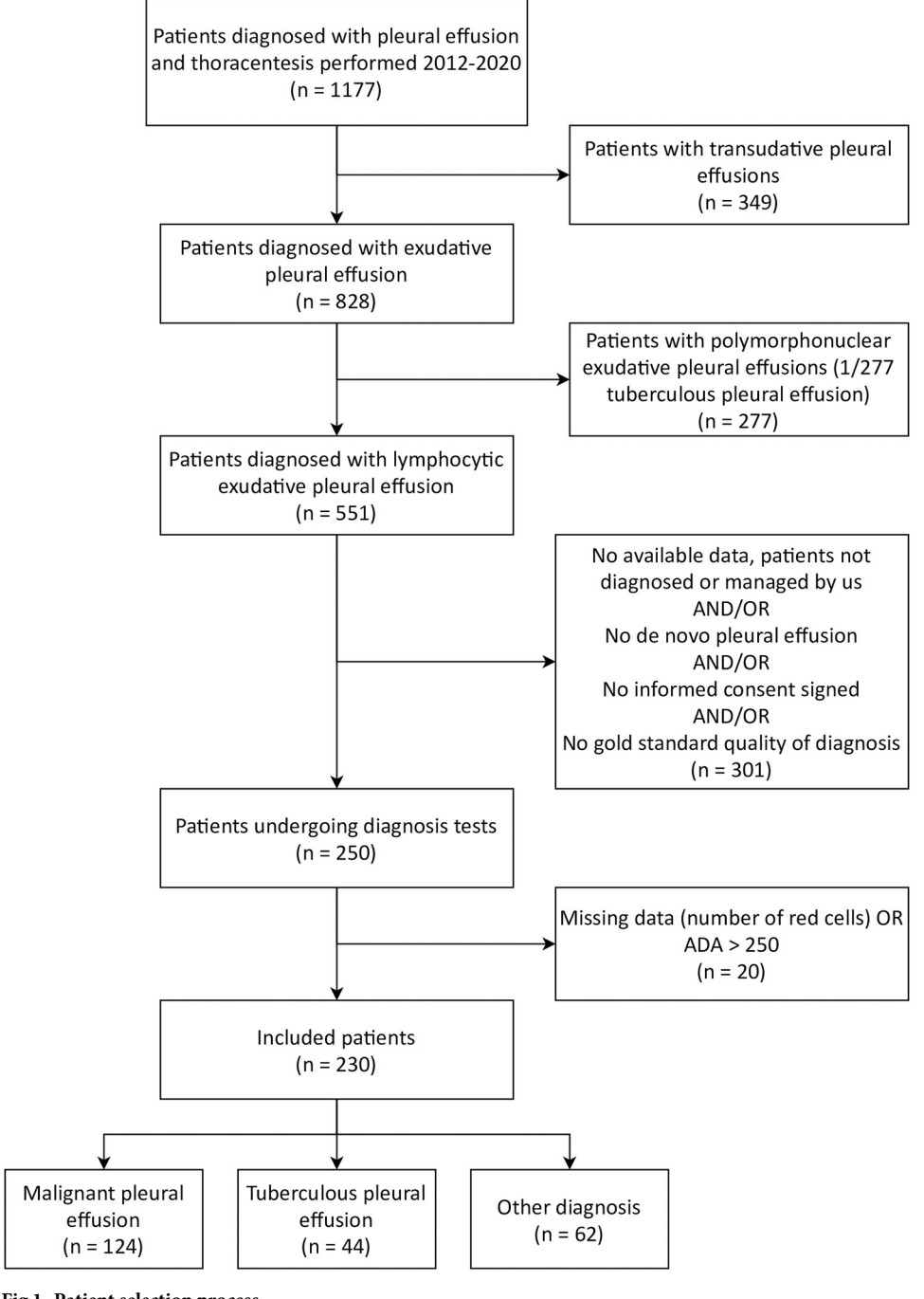

**Fig 1. Patient selection process.**

balanced accuracy and the one-vs-rest AUC, the last of these yielding the best parameterizations.

In the binary classification problem (tuberculous vs. non-tuberculous), the validation predictions were used to calculate receiver operating characteristic (ROC) [28] and precision-recall (PR) [29] curves in order to obtain decision thresholds that maximized a given metric. The Youden index [30] was used to select a good balance between sensitivity and specificity in

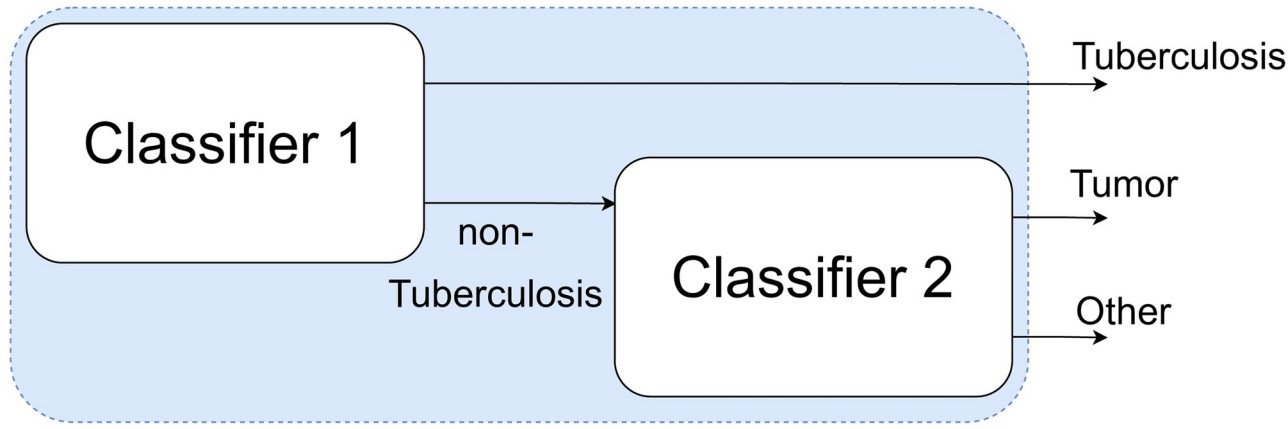

**Fig 2. Classification pipeline.** Classification pipeline.

the ROC curves. The F1 score was used in the PR curves. It is worth mentioning that these predictions were "clean", that is, they were obtained from the validation subset using a classifier trained with the rest of subsets within each fold of cross validation. These clean predictions were aggregated to calculate the validation results for each method.

In the three-class classification problem, the goal was to maximize the number of correct classifications of tuberculosis and MPE. Misclassification of samples of the class "Other" was considered a minor problem.

We applied two different approaches to accomplish this task. The first was a classification pipeline that used the previously trained binary classifier in combination with another classifier that distinguished between malignant effusions and the rest (Fig 2). Hence, a sample was analyzed in the first classifier, and if that classifier predicted tuberculosis, that was the final classification for that patient. Otherwise, the second classifier analyzed the sample to predict whether it corresponded to a tumor or some other disease. This second classifier was trained with the same training set as the first one, but not taking into consideration the samples corresponding to tuberculosis.

This first approach had the advantage that the thresholds of the two classifiers could be adjusted seeking to maximize the number of tuberculosis cases that were correctly predicted as such in the first classifier, as well as the number of tumors correctly classified in the second one.

The second approach to solve this problem was a multiclass classifier that distinguished at the same time between tuberculous, malignant and other effusions. Using a single classifier reduced the training effort but complicated the adjustment of the decision thresholds to make it possible to prioritize the prediction of tuberculosis and tumors, over the class "Other". In addition to all the aforementioned analysis, and considering the dynamics in our previous retrospective study, we also assessed the diagnostic accuracy of ADA level in this "lymphocytic exudative" scenario in terms of sensitivity, specificity, positive predictive value (PPV) and negative predictive value (NPV). Thereafter, we did the same, incorporating all the variables included in the machine learning algorithm. Our aim was to perform a Bayesian analysis of both sets of diagnostic variables.

## Results

Overall, 230 out of 1177 episodes of pleural effusion met all the criteria for being included in the study (de novo, exudative, lymphocytic, diagnosis according to the composite reference

standard, age over 16 years and written informed consent given). As shown in Table 1, the male:female ratio was 135/95 (58%/42%) and median age 69 years (1st and 3rd quartiles, 55 and 80 years respectively). Further, 12 out of 44 cases of tuberculous pleural effusion were diagnosed in non-Spanish patients (27%, p<0.001), all the clinical and laboratory parameters except cell count and pH differed significantly across the three groups: tuberculous, malignant, parapneumonic/other effusions (opting to merge these last two into a single group for subsequent analysis). Among all the samples, 44 patients were diagnosed with tuberculous pleural effusion (19.1%) and 124 (53.9%) with MPE—77 having lung cancer, 22 non-lung cancer, 10 lymphoma and 15 mesothelioma-, while 28 had parapneumonic effusions (12.1%) and 34 (14.7%) were diagnosed with other non-malignant diseases. All these data are set out in the Supporting Information section: S2 Table. Twenty-three of the 124 cases of MPE (18.5%) were classified as paramalignant: 19 of these patients having lung cancer, 2 non-lung cancer and 2 lymphomatous cancer.

The diagnostic yield of MTB culture, Xpert MTB/RIF and citology/histopathology both in pleural fluid as in pleural biopsy (closed or guided by thoracoscopy), along with the diagnostic yield of sputum, bronchoalveolar lavage and bronchial biopsy are included in the Supporting Information Section: S3 Table.

There were no cases of TPE with HIV infection in our series. Three patients previously diagnosed with HIV infection developed pleural effusions, in all cases malignant.

**Table 1. Characteristics of patients and pleural fluid samples by diagnosis.**

|  | Tuberculous | | Malignant | | Parapneumonic / Other effusions | | |
|---|---|---|---|---|---|---|---|
|  | N | % | N | % | N | % | p-value |
| Total | 44 | 19.1 | 124 | 53.9 | 62 | 27 |  |
| Sex |  |  |  |  |  |  | 0.21 |
| Male | 31 | 70.4 | 70 | 56.4 | 34 | 54.8 |  |
| Female | 13 | 29.6 | 54 | 43.6 | 28 | 45.2 |  |
| Nationality |  |  |  |  |  |  | <0.001 |
| Spanish | 32 | 72.7 | 121 | 97.6 | 58 | 93.6 |  |
| Other | 12 | 27.3 | 3 | 2.4 | 4 | 6.4 |  |
|  | Median | $P_{25} - P_{75}$ | Median | $P_{25} - P_{75}$ | Median | $P_{25} - P_{75}$ |  |
| Age, years[a] | 54.8 | 22.2 | 68.1 | 12.7 | 68.8 | 18.5 | 0.001 |
| ADA, U/l | 72 | 57.3–81 | 22 | 18–27.5 | 23 | 19–28 | <0.001 |
| pH | 7.4 | 7.4–7.5 | 7.4 | 7.4–7.5 | 7.5 | 7.4–7.5 | 0.08 |
| Glucose | 84.5 | 66–96 | 101 | 84–120.5 | 110.5 | 92–128 | <0.001 |
| Cell no. | 2315 | 1338.5–4126 | 1760.5 | 1140–2905 | 1715.5 | 618–3056 | 0.06 |
| MNC | 93 | 80–97.8 | 91.5 | 79.3–96 | 75 | 59–86 | <0.001 |
| PMNC | 7 | 2.2–20 | 8.5 | 4.05–20.75 | 25.5 | 14–41 | <0.001 |
| RBC | 3650 | 1805–10000 | 8000 | 2840–29300 | 6950 | 2700–40000 | 0.04 |
| LDH, U/l | 444.5 | 266.5–651.5 | 403 | 212–623 | 212.5 | 165–372 | <0.001 |
| Proteins | 5 | 4.6–5.3 | 4.4 | 3.8–4.8 | 4.3 | 3.5–4.9 | <0.001 |
| LDH / ADA | 6.1 | 4.2–10.7 | 15.5 | 9–28.4 | 10.1 | 7.3–15.9 | <0.001 |

ADA: adenosine deaminase; Cell no: number of cells/mm3; MNC: percentage of mononuclear cells in pleural fluid; PMNC: percentage of polymorphonuclear cells in pleural fluid; RBC: number of red blood cells/mm3. LDH: lactate dehydrogenase.

[a] Value shown as Mean and Standard deviation.

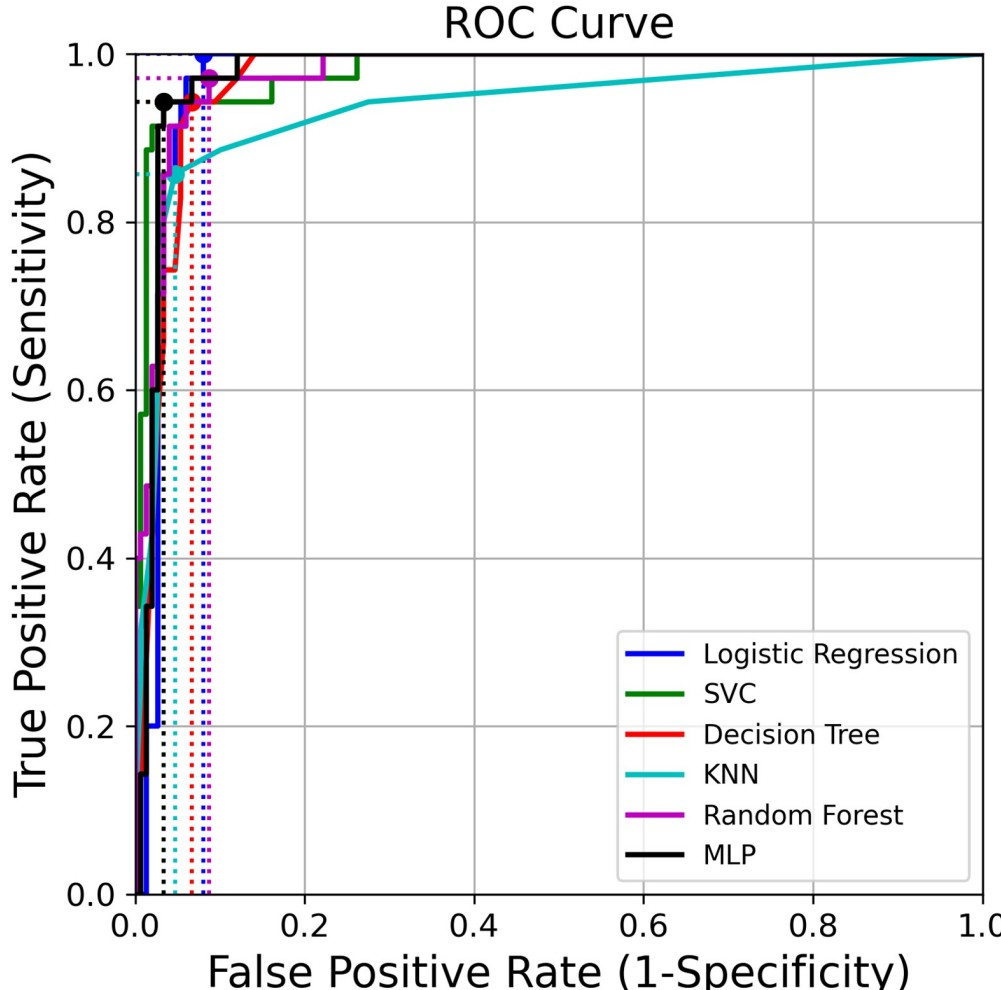

**Fig 3. Receiver operating characteristic curves using validation predictions.** The dots correspond to the points that maximize the Youden index.

## Classifying tuberculous vs. non-tuberculous effusions

Figs 3 and 4 show the ROC and PR curves for the binary classifier, respectively.

Table 2 lists performance scores for each method using three different thresholds: 0.5, the one that maximizes the Youden index on the ROC curve and the one that maximizes the F1 score on the PR curve. KNN was the classifier with the lowest value for the AUC (0.94); Logit and DT achieved an AUC of 0.97; whereas SVC, RF and MLP obtained 0.98.

The classifiers with the best set of parameters (S4 Table) and decision thresholds found were trained with the complete training set (184 samples). These classifiers were then tested using the 46 samples of the test set. Table 3 shows the scores for each method using the best threshold found in the validation stage.

In the validation stage, as can be seen in Table 2, the SVC achieved the best results: accuracy of 97%, AUC of 0.98, sensitivity of 91% and specificity of 98% (threshold of 0.35). In the testing phase (Table 3), however, the best results were obtained with the logistic regression: accuracy of 96%, AUC of 0.96, sensitivity of 100% and specificity of 95% (threshold of 0.28).

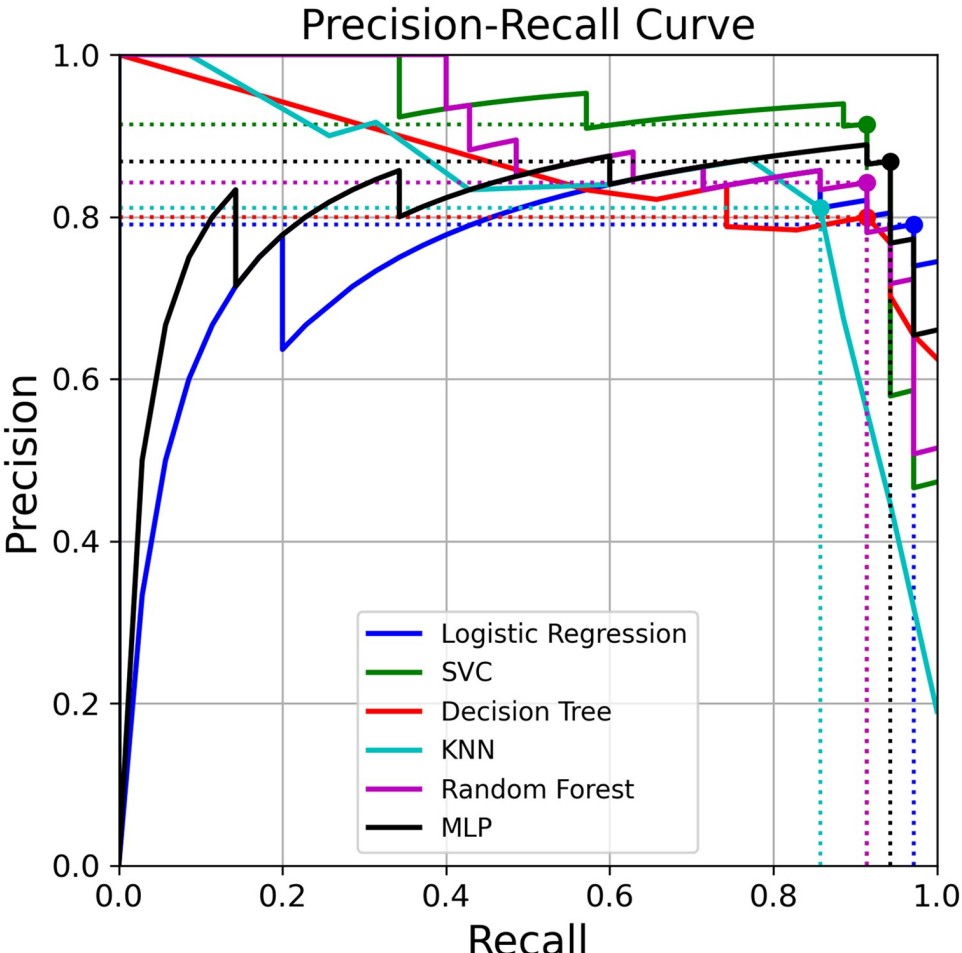

**Fig 4. Precision-recall curves using validation predictions.** Precision-recall curve for each method using the validation predictions. The dots correspond to the points that maximize the F1 score.

## Classifying tuberculous vs. malignant vs. other effusions

The first approach to performing a three-fold classification used the previous binary classifier for its first stage. The results obtained for the second stage of the pipeline are provided in the Supporting information section: S1 and S2 Figs, S5 and S6 Tables. Considering the validation results for each of the binary classifiers of the pipeline, we decided to use the SVC in both stages.

The pipeline achieved an overall accuracy of 79% in the validation predictions and 80% in the test set. To determine how the three-class classifier behaved in terms of sensitivity and specificity in the two-class problem (Tuberculous and Other), we have combined the results of the classes "Malignant" and "Other" into a single class. By doing this, the sensitivity and specificity would remain the same in the Tuberculous/Other classification, but this type of classifier would also allow us to detect 90% of malignant cases.

The second approach to solve the multiclass classification problem was to train, from scratch, a classifier that could distinguish between tuberculous, malignant and other effusions. The scores obtained with these classifiers are reported in the Supporting Information section: S7 and S8 Tables.

**Table 2. Accuracy (Acc), sensitivity (SEN), specificity (SPF) and F1 score (F1) of all the classifiers in the validation stage, using three different thresholds (T): 0.5, the one that maximizes the Youden index on the receiver operating characteristic curve and the one that maximizes the F1 score on the precision-recall curve, and their confidence intervals at 95% level.**

| Method | T | AUC | Acc (95% CI) | SEN (95% CI) | SPF (95% CI) | F1 (95% CI) |
|---|---|---|---|---|---|---|
| | | | **Threshold: 0.5** | | | |
| Logit | 0.5 | 0.97 | 0.92 (0.87,0.95) | 0.71 (0.54, 0.85) | 0.97 (0.92,0.99) | 0.77 (0.65, 0.87) |
| SVC | 0.5 | 0.98 | 0.97 (0.93,0.99) | 0.89 (0.73,0.97) | 0.99 (0.95,1.00) | 0.91 (0.83, 0.97) |
| DT | 0.5 | 0.97 | 0.92 (0.88, 0.96) | 0.83 (0.66,0.93) | 0.95 (0.90,0.98) | 0.81 (0.69, 0.90) |
| KNN | 0.5 | 0.94 | 0.91 (0.86,0.95) | 0.66 (0.48, 0.81) | 0.97 (0.93,0.99) | 0.74 (0.60, 0.86) |
| RF | 0.5 | 0.98 | 0.94 (0.90,0.97) | 0.83 (0.66,0.93) | 0.97 (0.92,0.99) | 0.84 (0.73, 0.92) |
| MLP | 0.5 | 0.98 | 0.96 (0.92,0.98) | 0.89 (0.73,0.97) | 0.97 (0.93,0.99) | 0.89 (0.80, 0.96) |
| | | | **Threshold: max Youden Index** | | | |
| Logit | 0.26 | 0.97 | 0.93 (0.89, 0.97) | 1.00 (0.90,1) | 0.92 (0.86,0.96) | 0.85 (0.76, 0.92) |
| SVC | 0.29 | 0.98 | 0.96 (0.92, 0.98) | 0.94 (0.81,0.99) | 0.97 (0.92,0.99) | 0.90 (0.82, 0.97) |
| DT | 0.25 | 0.97 | 0.93 (0.89, 0.97) | 0.94 (0.81,0.99) | 0.93 (0.88,0.97) | 0.85 (0.75, 0.92) |
| KNN | 0.30 | 0.94 | 0.93 (0.89,0.97) | 0.86 (0.70,0.95) | 0.95 (0.91,0.98) | 0.83 (0.72, 0.92) |
| RF | 0.19 | 0.98 | 0.92 (0.88,0.96) | 0.97 (0.85,1.00) | 0.91 (0.86,0.95) | 0.83 (0.73, 0.91) |
| MLP | 0.17 | 0.98 | 0.96 (0.92,0.98) | 0.94 (0.81,0.99) | 0.97 (0.92,0.99) | 0.90 (0.82, 0.97) |
| | | | **Threshold: max F1 score** | | | |
| Logit | 0.28 | 0.97 | 0.95 (0.90,0.97) | 0.97 (0.85,1.00) | 0.94 (0.89,0.97) | 0.87 (0.78, 0.94) |
| SVC | 0.35 | 0.98 | 0.97 (0.93,0.99) | 0.91 (0.77,0.98) | 0.98 (0.94,1.00) | 0.91 (0.84, 0.97) |
| DT | 0.40 | 0.97 | 0.94 (0.90,0.97) | 0.91 (0.77,0.98) | 0.95 (0.90,0.98) | 0.85 (0.76, 0.93) |
| KNN | 0.30 | 0.94 | 0.93 (0.89,0.97) | 0.86 (0.70,0.95) | 0.95 (0.91,0.98) | 0.83 (0.72, 0.92) |
| RF | 0.43 | 0.98 | 0.95 (0.91,0.98) | 0.91 (0.77, 0.98) | 0.96 (0.91,0.99) | 0.88 (0.78,0.95) |
| MLP | 0.17 | 0.98 | 0.96 (0.92,0.98) | 0.94 (0.81, 0.99) | 0.97 (0.92,0.99) | 0.90 (0.82, 0.97) |

None of these classifiers outperforms the combination of classifiers from the previous section in the validation predictions. As can be seen in Table S6 Table, SVC and decision tree classifiers obtain the best results in terms of accuracy (76% and 74%, respectively). The decision tree achieved a sensitivity of 91% and specificity of 94% in the Tuberculous/Other classification.

As a result, in the validation predictions, we were able to classify correctly 32 out of 35 cases of TPE and 89 out of 98 cases of MPE (sensitivity of 91.4% for TPE and 90.8% for MPE); conversely, a total of 3 additional cases were mistakenly classified as TPE out of 149 non-TPE cases and 29 cases were mistakenly classified as MPE out of 86 non-MPE cases (specificity of 98.0% for TPE and 66.3% for MPE). Furthermore, all 3 cases of TPE misdiagnosed were classified as MPE, theoretically leading to additional diagnostic procedures, and only 6 out of the 98

**Table 3. Threshold (T), area under the curve (AUC), accuracy (Acc), sensitivity (SEN), specificity(SPF) and F1 score (F1) of all the classifiers in the test stage, using the best thresholds found in the validation stage with their confidence intervals at 95% level.**

| Method | T | AUC | Acc (95% CI) | SEN (95% CI) | SPF (95% CI) | F1 (95% CI) |
|---|---|---|---|---|---|---|
| Logit | 0.28 | 0.98 | 0.96 (0.85,0.99) | 1.00 (0.66,1.00) | 0.95 (0.82,0.99) | 0.90 (0.71,1.00) |
| SVC | 0.35 | 0.96 | 0.93 (0.82,0.99) | 0.89 (0.52,1.00) | 0.95 (0.82,0.99) | 0.84 (0.60,1.00) |
| DT | 0.40 | 0.96 | 0.96 (0.85,0.99) | 1.00 (0.66,1.00) | 0.95 (0.82,0.99) | 0.90 (0.71,1.00) |
| KNN | 0.30 | 0.96 | 0.91 (0.79,0.98) | 0.78 (0.40,0.97) | 0.95 (0.82,0.99) | 0.78 (0.48,0.96) |
| RF | 0.43 | 0.98 | 0.93 (0.82,0.99) | 0.89 (0.52,1.00) | 0.95 (0.82,0.99) | 0.84 (0.63,1.00) |
| MLP | 0.17 | 0.98 | 0.93 (0.82,0.99) | 0.89 (0.52,1.00) | 0.95 (0.82,0.99) | 0.84 (0.59,1.00) |

cases of MPE were misdiagnosed as "Other", that is, 93.9% of cases of MPE (92/98) were classified as MPE or TPE. The relatively low specificity of the classification for MPE was mainly due to overdiagnosis of MPE amongst cases in the other diagnosis group. Indeed, we deliberately implemented the classification into the three groups prioritizing the overdiagnosis of MPE over its underdiagnosis.

In the test set, we were able to correctly classify 8 out of 9 cases of TPE and 24 out of 26 cases of MPE (sensitivity of 88.8% for TPE and 92.3% for MPE); conversely, a total of 2 additional cases were mistakenly classified as TPE out of 37 non-TPE cases and 6 cases were mistakenly classified as MPE out of 20 non-MPE cases (specificity of 94.5% for TPE and 70.0% for MPE). Only 1 of the 26 cases of MPE was misdiagnosed as "Other", that is, 96.1% of cases of MPE (25/26) were classified as MPE or TPE.

## Performance of the classifiers versus ADA alone

We compared the results of our machine-learning-based classifiers with those derived from using ADA as the only diagnostic criteria. We obtained three thresholds for ADA: default threshold (in this case 40 U/l), the threshold that maximized the Youden index in the ROC curve, and the threshold that maximized the f1-score in the PR curve. It is interesting to note that the threshold that maximized the Youden index with our samples was 40 U/l as well. The threshold that maximized the f1 score in the PR curve was 43 U/l. With these thresholds two set of metrics were extracted:

- Threshold = 40.00: Accuracy 0.93, Sensitivity 0.97 and Specificity 0.93

- Threshold = 43.00: Accuracy 0.95, Sensitivity 0.94 and Specificity 0.95

Our classifiers obtained better Sensitivity/Specificity values. For example, Logistic Regression obtained 0,97/0,94 (youden 0.91). SVC and MLP gets 0,94/0,97 (youden 0.91). Also, better Accuracy is obtained with SVC (0,97) and MLP (0,96).

Regarding the Bayesian analysis of the diagnostic accuracy of two sets of diagnostic variables and with a pre-test probability of 3.8%:

- for ADA >40 U/l (plus lymphocyte percentage > 50%), sensitivity, specificity, PPV and NPV were 97%, 93%, 35% and 100% respectively.

- for the whole set of variables included in the machine learning algorithms, sensitivity, specificity, PPV and NPV were 91%, 98%, 64% and 100% respectively.

Pre- and post-test probabilities of both diagnostic options are shown in Fig 5. Additionally, Table 4 lists the aforementioned parameters with respect to the corresponding pre-test probability up to 50%. Notably, for a pre-test probability of 10% (the upper limit of low prevalence) and using the full set of variables included in the machine learning algorithms, the PPV already reaches 83.5% and NPV remains at 99%.

## Discussion

Our study has prospectively analyzed the performance of non-invasive and straightforward diagnostic methods combining ADA level with routine parameters in pleural fluid and age, in the diagnosis of TPE using a machine learning approach, in a setting with a low-to-intermediate incidence of general TB (14 cases/100,000 population/year) and low prevalence scenario for TPE (3.8%). Secondly, we have sought to distinguish among non-tuberculous effusions those which are most likely to be malignant.

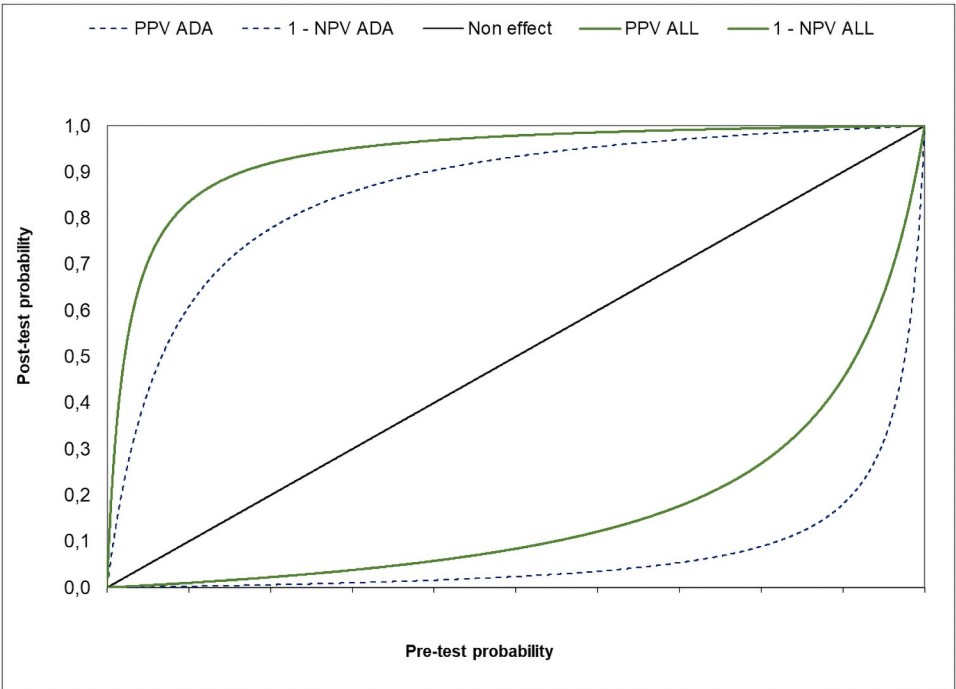

**Fig 5. Post-test probability of TPE after positive (top) or negative (bottom) results of 1) ADA>40 U/l plus implicit lymphocyte percentage >50% alone or 2) in addition to age and routine pleural fluid parameters included in the machine learning algorithms; for different pre-test probabilities of disease.**

The definitive diagnosis of TPE remains a challenge due to its paucibacillary nature, and many attempts have been made to find an alternative method to the more invasive and hazardous pleural biopsy. Biomarkers, rapid cultures and PCR-based techniques have been the main focus of research over recent decades [31–33]. Measurement of pleural fluid ADA has shown a uniformly high diagnostic performance for TPE in five major consecutive meta-analyses with a mean sensitivity and specificity of 92–93% and 90–92% respectively for a diagnostic threshold of >40 U/l [34–38].

**Table 4. Comparative table of Bayesian probabilities of test parameters used: ADA > 40 U/l (plus lymphocyte percentage > 50%) versus the whole set of variables included in the machine learning algorithms.** PPV: positive predictive value. NPV: negative predictive value.

| | ADA + LYM | | All | |
|---|---|---|---|---|
| Sensitivity | 98% | | 91% | |
| Specificity | 93% | | 98% | |
| Pre-test probability | PPV | NPV | PPV | NPV |
| 5% | 42.4% | 99.9% | 70.5% | 99.5% |
| 10% | 60.8% | 99.7% | 83.5% | 99.0% |
| 15% | 71.2% | 99.6% | 88.9% | 98.4% |
| 20% | 77.8% | 99.4% | 91.9% | 97.8% |
| 30% | 85.7% | 99.0% | 95.1% | 96.2% |
| 40% | 90.3% | 98.4% | 96.8% | 94.2% |
| 50% | 93.3% | 97.6% | 97.8% | 91.6% |

Following the Bayesian interpretation of its diagnostic accuracy, this biomarker is optimal as a rule-in test in high TB prevalence settings, and it is currently considered useful as a rule-out test in low prevalence scenarios [5, 31, 32]. Many other biomarkers have also been extensively studied for the same purpose, unstimulated interferon-gamma (best sensitivity of 89% and specificity of 97%) [39, 40] and interleukin-27 (best sensitivity of 96% and specificity of 99%) [41–43] being the most accurate, but they lack the historical success, low cost, availability and simplicity of ADA measurement [39, 44]. Furthermore, concerns regarding the accuracy of ADA in immunocompromised patients have been extensively allayed: it retains a high utility for the diagnosis of TPE in all HIV-infected patients, even those with low CD4 counts [45, 46] and in renal transplant recipients [47].

The etiological spectrum of pleural effusions obviously depends on the population studied: in developing countries, tuberculosis and bacterial infections are by far the commonest causes; conversely, in developed countries, more than 90% of pleural effusions are caused by congestive heart failure, malignancy, pneumonia and pulmonary embolism [42, 48]. In this sense, the pre-test probability of a disease decisively affects the value of a given diagnostic tool, but the combination of various tests or parameters could theoretically improve the post-test probability of a given suspected disease, as has been found in our retrospective study and others [8, 49, 50]. Specifically, in 2012, we were able to demonstrate that the addition of lymphocyte percentage >50% to ADA>40 U/l in pleural fluid increased the specificity and the PPV of the latter, mostly in low-to-intermediate incidence scenarios. Some other authors have reported their experience with ADA subsequently in intermediate-low [51] and low [52–54] incidence settings, with promising results.

In our current study, we expanded the number of parameters to be combined with ADA, and we performed a statistical analysis through the classification of all cases by a machine learning approach into three groups, our main goal being to correctly classify cases of TPE, and in a second step, we sought to also do the same with cases of MPE, even if these would be overdiagnosed. In relation to this, if we exclude parapneumonic effusions and acute inflammatory exudates, which are usually neutrophilic (even cases of TPE in the first 24–96 hours), most of the remaining types of pleural effusion are lymphocytic, suggesting non-acute disease, and differential diagnosis between TPE and MPE remains one of the most pressing challenges in this scenario [49, 55–60].

As a result, and in line with the data described above concerning the three-fold classification pipeline, in the validation predictions, we were able to classify those cases of TPE with a sensitivity and specificity of 91.4% and 98.0% respectively. In regard to MPE cases we obtained a sensitivity of 90.8% and a specificity of 66.2% (due to prioritization of overdiagnosis). Besides, the corresponding parameters were similar in the test set (sensitivity and specificity 88.8% and 94.5% for TPE; and 92.3% and 70.0% for MPE). We have been able to find three reports regarding diagnosis of pleural tuberculosis by artificial intelligence based diagnostic models [61–63]. All of them obtained a sensitivity and specificity over 90%, but some essential differences should be pointed out when comparing with ours: first and foremost, they are performed in very high prevalence scenarios (namely, China and Brazil, accounting for 96 out of 137, 73 out of 140 and 192 out of 443 patients diagnosed with TPE respectively); second, patients with clinical diagnosis of tuberculosis and improvement with antituberculous treatment are included, or even only microbiologically confirmed cases by culture are admitted in one of the studies [62].

Furthermore and regarding the Bayesian analysis of the accuracy of any parameter and/or diagnostic tool for the presumptive diagnosis of TPE, in this prospective study, we have been able to achieve even better results than in our previous retrospective study [8] with the combination of the full set of parameters included in the machine learning algorithm, all of them

easy to obtain in routine clinical practice: the addition of the routine parameters and age had the benefit of improving the specificity and PPV with respect to using ADA (plus implicit lymphocytic percentage >50% in all pleural fluids), in a low prevalence scenario. Notably, for a TPE prevalence of 3.8%, specificity and PPV increased from 93 to 98% and from 35% to 64% respectively (Fig 5).

Some additional results deserve to be highlighted:

- Pleural fluid culture yielded MTB in 25 out of 44 first samples (56%), and additionally, in 3 subsequent samples (63%), probably due to the use of the BACTEC MGIT semi-automated system, which is known to provide a higher and faster yield than solid media, with a reported sensitivity of as high as 63% in pleural fluid samples in general [64] and even 79% in samples from HIV-positive patients [65]. Pleural tissue sample culture yielded MTB from 7 out of 12 closed biopsies performed (58%) and 7 out of 9 thoracoscopic biopsies (77%).

- Regarding the performance of the Xpert MTB/RIF assay, in our study, it detected MTB genome in 5 out of 44 pleural fluid samples (11%), 6 out of 12 closed pleural biopsy specimens (50%) and 6 out of 9 pleural thoracoscopic biopsy specimens (66%) (combined sensitivity in pleural tissue samples of 57%). The specificity reached 100% and no cases of rifampin resistance were identified. The difference in terms of sensitivity between pleural fluid and pleural tissue samples is evident. Overall, the sensitivity of molecular analysis of pleural fluid remains insufficient. In a recent Cochrane analysis on the diagnostic accuracy of the Xpert MTB/RIF assay for extrapulmonary tuberculosis, only five studies were found including pleural fluid and the composite reference standard, these reporting a sensitivity of 13–29% and specificity of 97–100% [66]. This low sensitivity might be due to small numbers of bacilli or even related to local intrinsic inhibitors or intracellular sequestration of mycobacteria [31].
The novel next-generation Xpert MTB/RIF Ultra lowers the detection limit from 112.6 CFUs/mL to 15.6 CFUs/mL and shows promising results: in a recent multicenter study, it showed a sensitivity of 44% and specificity of 98% in pleural fluid compared to the composite reference standard, [67]. Data regarding the performance of this assay in pleural tissue samples and against the composite reference standard is scarce: the aforementioned Cochrane analysis only found a single study on this topic including 55 patients, 14 out of them with tuberculous effusions, and with surprising and disappointing results (sensitivity of 0%, specificity of 98%) [66]. Subsequently, three additional studies of 73, 113 and 27 patients diagnosed with TPE and who underwent thoracoscopic pleural biopsy reported a sensitivity of the Xpert MTB/RIF assay of 45%, 69% and 85%, respectively in pleural tissue samples against the composite reference standard [68–70]. Our study showed a five-fold higher sensitivity of the Xpert MTB/RIF assay in pleural tissue compared to pleural fluid (57% versus 11%), and close to that obtained with culture of pleural fluid (56%) or closed biopsy pleural specimens (58%). Despite showing low sensitivity, PCR-based techniques in pleural fluid are currently considered useful because they are not invasive and their specificity is universally optimum; additionally, the Xpert MTB/RIF assay detects rifampicin resistance with sensitivity and specificity of 95% and 98.8% respectively [32]. On the other hand, the performance of the Xpert MTB/RIF assay in pleural tissue samples is close to that of MTB culture, making it, in our opinion, a diagnostic tool that should be considered.

Our study does have some limitations: only patients with lymphocytic exudative pleural effusions were included in this prospective study. This decision was influenced by financial limitations and guided by the fact that virtually all tuberculous pleural effusions become lymphocytic [17–19], and that the differential diagnosis between TPE and MPE remains a great

concern in just this scenario. As noted above, only one additional patient diagnosed with TPE showed a first pleural fluid sample that was neutrophilic, and this underlines the value of our results. The relatively limited number of patients is mainly due to the low prevalence of the tuberculous disease in our clinical setting; nevertheless, statistical power is guaranteed and enough as we expressed in the Material and Methods section. On the other hand, the main strengths of our study lie in its prospective and multicenter nature, along with the certainty of the diagnosis in all patients included; in this regard, the homogeneity of well differentiated populations makes the use of Machine Learning feasible with fewer patients.

## Conclusion

In conclusion, ADA combined with age and routine pleural fluid parameters in a machine learning approach is suitable for the diagnosis of tuberculous pleural effusion in a low prevalence area. Secondly, discrimination (at the expense of overdiagnosis) of cases of malignant pleural effusion is an added benefit in this scenario.

## Supporting information

**S1 Fig. Receiver operating characteristic curve for each method using the validation predictions of the binary classification problem where the positive class is malignant and the negative class is other.** All training samples have been used except the ones identified as Tuberculous. The dots correspond to the points that maximize the Youden index.
(TIF)

**S2 Fig. Precision-recall curve for each method using the validation predictions of the binary classification problem where the positive class is *malignant* and the negative class is *other*.** All training samples have been used except the ones identified as Tuberculous. The dots correspond to the points that maximize the F1 score.
(TIF)

**S1 Table. Relative importance of features in the classification process.** The Random Forest classifier allows estimating the relative importance of each variable in the classification process. In view of the results, and in accordance with all the previously reported studies, in this classification process, "ADA" is the most important variable, followed by "age". The rest of the variables are less useful, but we decided not to remove any of them a priori, but rather let each Machine Learning method assign a relative importance to each of them.
(PDF)

**S2 Table. Classification of patients by etiology.**
(PDF)

**S3 Table. Methods and comparative diagnostic yield regarding tuberculous and malignant pleural effusions.**
(PDF)

**S4 Table. Parameters of the classifiers.** Logistic Regression (Logit). Support Vector Machine (SVC). Decision Tree (DT). K-Nearest Neighbors (KNN). Multi Layer Perceptron (MLP).
(PDF)

**S5 Table. Validation results of the binary classification problem where the positive class is *malignant* and the negative class is *other*.** All training samples have been used except the ones identified as Tuberculous. Area under the curve (AUC), accuracy (Acc), sensitivity (SEN), specificity (SPF) and F1 score (F1) of all the classifiers using three different thresholds:

0.5, the one that maximizes the Youden index in the receiver operating characteristic curve and the one that maximizes the F1 score in the precision-recall curve.
(PDF)

**S6 Table. Test results of the binary classification problem where the positive class is *malignant* and the negative class is *other*.** All test samples have been used except the ones identified as Tuberculous. Threshold (T), area under the curve (AUC), accuracy (Acc), sensitivity (SEN), specificity(SPF) and F1 score (F1) of all the classifiers, using the best thresholds found in the validation stage.
(PDF)

**S7 Table. Multiclass classifier scores (Tuberculous, malignant and other).** Accuracy (Acc), balanced accuracy (bAcc), weighted F1 score (wF1) and Area Under the Curve using one-vs-rest strategy (AUC) of all classifiers using the validation samples.
(PDF)

**S8 Table. Multiclass classifier scores (Tuberculous, malignant and other).** Accuracy (Acc), balanced accuracy (bAcc), weighted F1 score (wF1) and area under the curve using one-vs-rest strategy (AUC) of all classifiers using the test samples.
(PDF)

## Acknowledgments

Lead author of Consortium: Cilla C M.D. Ph. D. Microbiology Service. Donostia University Hospital. CARLOSGUSTAVOSANTIAGO.CILLAEGUILUZ@osakidetza.eus

Cuñado A. Internal Medicine Service. Mendaro Hospital. AMAIA.CUNADOEIZAGUIRRE@osakidetza.eus, Moreno A. Internal Medicine Service. Mendaro Hospital. ANA.MORENORODRIGO@osakidetza.eus, Maiz A. Internal Medicine Service. Mendaro Hospital. ARANZAZU.MAIZEGANA@osakidetza.eus, Azcune A. Infectious Diseases Unit. Donostia University Hospital. ARKAITZ.AZCUNEGALPARSORO@osakidetza.eus, Labeguerie B. Pneumology Service. Donostia University Hospital. BENAT.LABEGUERIEARENAZA@osakidetza.eus, De la Guerra C. Internal Medicine Service. Mendaro Hospital. CARLA.DELAGUERRAACEBAL@osakidetza.eus, Martinez C. Internal Medicine Service. Mendaro Hospital. CINTIAMARIA.MARTINEZMATEU@osakidetza.eus, Sanchez E. Internal Medicine Service. Policlínica Gipuzkoa. Eloyyolanda@gmail.com, Montero E. Internal Medicine Service. Mendaro Hospital. ESPERANZA.MONTEROAPARICIO@osakidetza.eus, Michel FJ. Pneumolgy Service. Donostia University Hospital. FRANCISCOJAVIER.MICHELDELAROSA@osakidetza.eus, Acosta F. Internal Medicine Service. Mendaro Hospital. FEDERICORAMON.ACOSTAMAESTRE@osakidetza.eus, Bonache F. Internal Medicine Service. Mendaro Hospital. Fran.bonache@gmail.com, Urcelay G. Biochemistry Laboratory. Mendaro Hospital. MARIAGORETTI.URCELAYZALDUA@osakidetza.eus, De los Santos I. Pneumology Service. Mendaro Hospital. IDANIA.DELOSSANTOSVENTURA@osakidetza.eus, Perez I. Pneumology Service. Donostia University Hospital. IDOIA.PEREZSAMPEDRO@osakidetza.eus, Salegui I. Pneumology Service. Donostia University Hospital. INAKI.SALEGIETXEBESTE@osakidetza.eus, Sayago I. Pneumology Service. Asuncion Clinic. itxasos@telefonica.net, Zabaleta J. Thoracic Surgery Service. Donostia University Hospital. JON.ZABALETAJIMENEZ@osakidetza.eus, Iribarren JA. Infectious Diseases Unit. Donostia University Hospital. JOSEANTONIO.IRIBARRENLOYARTE@osakidetza.eus, Royo JI. Pneumology Service. Donostia University Hospital. JOSEIGNACIO.ROYOGUTIERREZ@osakidetza.eus, Garces JL. Internal Medicine Service. Policlinica Gipuzkoa. dr.jlgarces@gmail.com, Vaquero JM. Internal

Medicine Service. Mendaro Hospital. JOSEMARIA.VAQUEROHERNANDEZ@osakidetza.eus, Izquierdo JM. Thoracic Surgery Service. Donostia University Hospital. JOSEMIGUEL.IZQUIERDOELENA@osakidetza.eus, Mendiola I. Microbiology Service. Mendaro Hospital. JOSUNE.MENDIOLAARZA@osakidetza.eus, Miguel JA. Pneumology Service. Bidasoa Hospital. JUANANTONIO.MIGUELARCE@osakidetza.eus, Merino JL. Internal Medicine Service. Policlinica Gipuzkoa. juanluismerino@msn.com, De Juan MD. Immunology Service. Donostia University Hospital. MARIADOLORES.DEJUANECHAVARRI@osakidetza.eus, Alvarez M. Internal Medicine Service. Mendaro Hospital. MARIA.ALVAREZDECASTRO@osakidetza.eus, Temprano M. Pneumology Service. Goi Deba Hospital. MIKEL.TEMPRANOGOGENOLA@osakidetza.eus, Zubeltzu M. Pneumology Service. Donostia Universitary Hospital. MIKEL.ZUBELTZUSESE@osakidetza.eus, Azcue N. Microbiology Service. Donostia Universitary Hospital. MIRENNEKANE.AZCUEALBIZU@osakidetza.eus, De Andres N. Internal Medicine Service. Mendaro Hospital. NEREA.ANDRESIMAZ@osakidetza.eus, Casanova Y. Internal Medicine Service. Policlinica Gipuzkoa. ycasanov4@gmail.com, Bernardo P. Internal Medicine Service. Mendaro Hospital. MARIAPILAR.BERNARDOGALAN@osakidetza.eus, Fuentes S. Pneumology Service. Mendaro Hospital. SHANDRA.FUENTESPICADO@osakidetza.eus, Dorronsoro S. Pneumology Service. Urola Hospital. SILVIA.DORRONSOROQUINTANA@osakidetza.eus, Chic S. Pneumology Service. Mendaro Hospital. SUSANA.CHICPALACIN@osakidetza.eus, Camino X. Infectious Diseases Unit. Donostia University Hospital. XABIER.CAMINOORTIZDEBARRON@osakidetza.eus.

## Author Contributions

**Conceptualization:** Alberto Garcia-Zamalloa, Jorge Taboada, Emilio Perez-Trallero.

**Data curation:** Alberto Garcia-Zamalloa, Diego Vicente, Arantzazu Arrospide, Urko Aguirre, Nekane Múgica, Ladislao Aldama, Borja Aguinagalde, Montserrat Jimenez, Edurne Bikuña, Miren Begoña Basauri, Marta Alonso, Emilio Perez-Trallero.

**Formal analysis:** Rafael Arnay, Arantzazu Arrospide, Jorge Taboada, Iván Castilla-Rodríguez, Urko Aguirre, Ladislao Aldama, Borja Aguinagalde, Miren Begoña Basauri, Marta Alonso.

**Funding acquisition:** Diego Vicente.

**Investigation:** Alberto Garcia-Zamalloa, Diego Vicente, Arantzazu Arrospide, Iván Castilla-Rodríguez, Urko Aguirre, Nekane Múgica, Ladislao Aldama, Borja Aguinagalde, Edurne Bikuña, Miren Begoña Basauri, Emilio Perez-Trallero.

**Methodology:** Rafael Arnay, Arantzazu Arrospide, Jorge Taboada, Iván Castilla-Rodríguez, Urko Aguirre.

**Project administration:** Diego Vicente.

**Resources:** Diego Vicente, Nekane Múgica.

**Software:** Rafael Arnay, Arantzazu Arrospide, Jorge Taboada, Iván Castilla-Rodríguez, Urko Aguirre.

**Supervision:** Alberto Garcia-Zamalloa, Montserrat Jimenez, Edurne Bikuña, Miren Begoña Basauri, Marta Alonso, Emilio Perez-Trallero.

**Validation:** Rafael Arnay, Arantzazu Arrospide, Jorge Taboada, Iván Castilla-Rodríguez, Urko Aguirre, Miren Begoña Basauri, Marta Alonso.

**Visualization:** Nekane Múgica, Ladislao Aldama, Borja Aguinagalde, Montserrat Jimenez.

**Writing – original draft:** Alberto Garcia-Zamalloa, Rafael Arnay.

**Writing – review & editing:** Alberto Garcia-Zamalloa, Rafael Arnay.

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
