## [Decision Letter · Decision Letter 0]

17 May 2021

PONE-D-21-13738

Diagnostic accuracy of adenosine deaminase for pleural tuberculosis in a low prevalence setting: a machine learning approach within a 7-year prospective multi-center study

PLOS ONE

Dear Dr. Garcia-Zamalloa,

Thank you for submitting your manuscript to PLOS ONE. After careful consideration, we feel that it has merit but does not fully meet PLOS ONE’s publication criteria as it currently stands. Therefore, we invite you to submit a revised version of the manuscript that addresses the points raised during the review process.

Please submit your revised manuscript. If you will need significantly more time to complete your revisions, please reply to this message or contact the journal office at plosone@plos.org. Please include the following items when submitting your revised manuscript:

We look forward to receiving your revised manuscript.

Kind regards,

Frederick Quinn

Academic Editor

PLOS ONE

Journal Requirements:

2) You indicated that you had ethical approval for your study. In your Methods section, please ensure you have also stated whether you obtained consent from parents or guardians of the minors included in the study or whether the research ethics committee or IRB specifically waived the need for their consent.

3) One of the noted authors is a group or consortium [Gipuzkoa Pleura Group Consortium]. In addition to naming the author group, please list the individual authors and affiliations within this group in the acknowledgments section of your manuscript. Please also indicate clearly a lead author for this group along with a contact email address.

Reviewers' comments:

Reviewer's Responses to Questions

**Comments to the Author**

1. Is the manuscript technically sound, and do the data support the conclusions?

Reviewer #1: Partly

Reviewer #2: Partly

2. Has the statistical analysis been performed appropriately and rigorously? 

Reviewer #1: Yes

Reviewer #2: I Don't Know

3. Have the authors made all data underlying the findings in their manuscript fully available?

Reviewer #1: Yes

Reviewer #2: No

4. Is the manuscript presented in an intelligible fashion and written in standard English?

Reviewer #1: Yes

Reviewer #2: Yes

5. Review Comments to the Author

Reviewer #1: In this study, the authors investigated the diagnostic value of routine parameters of pleural effusion for tuberculosis pleurisy. Machine learning approaches were used to combine these parameters. I have some comments.

(1) Introduction: the definition of machine learning should be introduced.

(2) The manuscript should be reported following the STARD guideline.

(3) A flowchart should be used to depict the patient selection process.

(4) The discussion section should be shortened, and the readability needs to be improved.

(5) Two studies have investigated the clinical utility of machine learning approaches for TPE diagnostics (Respir Res. 2019 Oct 16;20(1):220.; Comput Methods Programs Biomed. 2018 Jan;153:211-225.). These two studies should be mentioned, and the strengthes of the current study should be discussed.

(6) Parameters used in machine learning algorithms should be reported.

(7) 95%CI should be added to each table.

(8) Line 348: is this a retrospective study?

(9) The term “validation dataset” should be modified as “training dataset.”

(10) The major limitation of this study is the small sample size.

(11) Methods: the software and packages used for machine learning should be reported.

Reviewer #2: The title is inappropriate as several other parameters besides ADA were used to train the algorithm for machine learning.

The notion that pretest probability remained 3.8% during the entire study period is incorrect. The denominator for this assumption includes all pleural effusions, while authors studied only lymphocytic exudative effusions.

During the study period, the entire province had only 45 patients of tuberculous pleural effusion. 44 of these were included in the present study. This appears to be a biased sample since all the lymphocytic exudative pleural effusions in the province don’t appear to have been included.

Patients where no definitive diagnosis could be reached were excluded, as were patients with any disease that could clearly explain the effusion. This should not have been done. Rather they could have used as the validation set for checking the accuracy of the algorithm.

I am still not sure whether the complicated machine learning algorithm actually improved diagnostic accuracy or not. Firstly, there is no clarity on why a particular set of variables were selected to train the algorithm. It is quite possible that less or more may have been better. One cannot decipher the contribution of each parameter towards the overall decision-making. In particular, the overall sensitivity of the algorithm for lymphocytic exudates was lower than that of ADA, but the specificity was higher. The same result could have been easily obtained by shifting the arbitrarily defined ADA threshold of 40 U/L to another value. Authors need to describe the diagnostic accuracy of ADA at different thresholds. For a confirmatory test, high specificity and acceptable sensitivity are needed, and this can be easily achieved using ADA alone.

The meaning and interpretation of the figures on ROC curves, precision-recall curves, and confusion matrices is not clear to me, not it has been adequately explained. I am not an expert in machine learning techniques, and neither is most of the readership of this journal.

6. PLOS authors have the option to publish the peer review history of their article (what does this mean?). If published, this will include your full peer review and any attached files.

Reviewer #1: No

Reviewer #2: No

---

## [Author Response · Author response to Decision Letter 0]

20 Jul 2021

We have no comment in this regard.

---

## [Decision Letter · Decision Letter 1]

6 Sep 2021

PONE-D-21-13738R1

Diagnostic accuracy of adenosine deaminase for pleural tuberculosis in a low prevalence setting: a machine learning approach within a 7-year prospective multi-center study

PLOS ONE

Dear Dr. Garcia-Zamalloa,

Thank you for submitting your manuscript to PLOS ONE. After careful consideration, we feel that it has merit but does not fully meet PLOS ONE’s publication criteria as it currently stands. Therefore, we invite you to submit a revised version of the manuscript that addresses the points raised during the review process.

Please submit your revised manuscript. If you will need significantly more time to complete your revisions, please reply to this message or contact the journal office at plosone@plos.org. Please include the following items when submitting your revised manuscript:

We look forward to receiving your revised manuscript.

Kind regards,

Frederick Quinn

Academic Editor

PLOS ONE

Reviewers' comments:

Reviewer's Responses to Questions

**Comments to the Author**

1. If the authors have adequately addressed your comments raised in a previous round of review and you feel that this manuscript is now acceptable for publication, you may indicate that here to bypass the “Comments to the Author” section, enter your conflict of interest statement in the “Confidential to Editor” section, and submit your "Accept" recommendation.

Reviewer #1: All comments have been addressed

2. Is the manuscript technically sound, and do the data support the conclusions?

Reviewer #1: No

3. Has the statistical analysis been performed appropriately and rigorously? 

Reviewer #1: No

4. Have the authors made all data underlying the findings in their manuscript fully available?

Reviewer #1: No

5. Is the manuscript presented in an intelligible fashion and written in standard English?

Reviewer #1: No

6. Review Comments to the Author

Reviewer #1: In this study, the authors investigated the diagnostic accuracy of conventional biomarkers for TPE. The strength of this study is machine learning analysis, which has been proved to be a valuable method to improve the diagnostic accuracy of biomarkers. I have some comments.

1. The current version of the manuscript is too long. The readability of the manuscript is poor. I strongly suggest authors shorten the length of the manuscript to make it more clear and focus. The current version of the manuscript is entirely unacceptable.

2.Figures 5, 6, 7, 8, 9, and 10 should be deleted because sensitivity and specificity have been reported in Table 2 and Table 3. In machine learning analysis, it is meaningless to report the results of the training cohort.

3. AUC should be reported in table 2.

7. PLOS authors have the option to publish the peer review history of their article (what does this mean?). If published, this will include your full peer review and any attached files.

Reviewer #1: No

---

## [Author Response · Author response to Decision Letter 1]

29 Sep 2021

Reviewer #1: In this study, the authors investigated the diagnostic accuracy of conventional biomarkers for TPE. The strength of this study is machine learning analysis, which has been proved to be a valuable method to improve the diagnostic accuracy of biomarkers. I have some comments.

Thank you for your comments. Please find below a point by point reply.

1. The current version of the manuscript is too long. The readability of the manuscript is poor. I strongly suggest authors shorten the length of the manuscript to make it more clear and focus. The current version of the manuscript is entirely unacceptable.

We regret that the reviewer found the manuscript difficult to read. Due to the nature of the contribution of the manuscript and the scope of the journal, we are aiming at three different groups of potential readers: clinicians, clinicians with solid training in statistics, and mathematicians/statisticians. We have tried to achieve a delicate balance between the depth provided to technical details and the clarity required to make our contribution accessible to clinicians who may be only interested in the application. A thorough explanation of technical details is unavoidable both to enhance the transparency of the methods and results, and to allow replicability.

Despite of all previously mentioned, we have done our best to make the manuscript easier and shorter whenever possible. 

2.Figures 5, 6, 7, 8, 9, and 10 should be deleted because sensitivity and specificity have been reported in Table 2 and Table 3. In machine learning analysis, it is meaningless to report the results of the training cohort.

Thank you for your comment. Following your recommendation we have removed these figures. 

3. AUC should be reported in table 2.

Thank you for your comment. Following your recommendation we have included the area under the curve in Table 2.

---

## [Decision Letter · Decision Letter 2]

15 Oct 2021

Diagnostic accuracy of adenosine deaminase for pleural tuberculosis in a low prevalence setting: a machine learning approach within a 7-year prospective multi-center study

PONE-D-21-13738R2

Dear Dr. Garcia-Zamalloa,

We’re pleased to inform you that your manuscript has been judged scientifically suitable for publication and will be formally accepted for publication once it meets all outstanding technical requirements.

Kind regards,

Frederick Quinn

Academic Editor

PLOS ONE

Additional Editor Comments (optional):

Reviewers' comments:

Reviewer's Responses to Questions

**Comments to the Author**

1. If the authors have adequately addressed your comments raised in a previous round of review and you feel that this manuscript is now acceptable for publication, you may indicate that here to bypass the “Comments to the Author” section, enter your conflict of interest statement in the “Confidential to Editor” section, and submit your "Accept" recommendation.

Reviewer #1: All comments have been addressed

2. Is the manuscript technically sound, and do the data support the conclusions?

Reviewer #1: Yes

3. Has the statistical analysis been performed appropriately and rigorously? 

Reviewer #1: Yes

4. Have the authors made all data underlying the findings in their manuscript fully available?

Reviewer #1: Yes

5. Is the manuscript presented in an intelligible fashion and written in standard English?

Reviewer #1: Yes

6. Review Comments to the Author

Reviewer #1: The quality of the manuscript has been improved, and the current version is acceptable.

7. PLOS authors have the option to publish the peer review history of their article (what does this mean?). If published, this will include your full peer review and any attached files.

Reviewer #1: No

---

## [Editor Report · Acceptance letter]

22 Oct 2021

PONE-D-21-13738R2 

Diagnostic accuracy of adenosine deaminase for pleural tuberculosis in a low prevalence setting: a machine learning approach within a 7-year prospective multi-center study 

Dear Dr. Garcia-Zamalloa:

I'm pleased to inform you that your manuscript has been deemed suitable for publication in PLOS ONE. Congratulations! Your manuscript is now with our production department. 

Kind regards, 

on behalf of

Dr. Frederick Quinn 

Academic Editor

PLOS ONE